# Rhizobacteria Isolated from Saline Soil Induce Systemic Tolerance in Wheat (*Triticum aestivum* L.) against Salinity Stress

**Noshin Ilyas [1],\***, **Roomina Mazhar [1]**, **Humaira Yasmin [2]**, **Wajiha Khan [3]**, **Sumera Iqbal [4]**, **Hesham El Enshasy [5,6,7],\*** and **Daniel Joe Dailin [5,6]**

[1] Department of Botany, PMAS-Arid Agriculture University, Rawalpindi 46300, Pakistan; roominamazhar83@gmail.com

[2] Department of Bio-Sciences, COMSATS University, Islamabad 45550, Pakistan; humaira.yasmin@comsat.edu.pk

[3] Department of Biotechnology, COMSATS University Islamabad, Abbottabad Campus, Abbottabad 22010, Pakistan; wajihak@cuiatd.edu.pk

[4] Department of Botany, Lahore College for Women University, Lahore 54000, Pakistan; sumeraiqbal2@yahoo.com

[5] Institute of Bioproduct Development (IBD), Universiti Teknologi Malaysia (UTM), Skudai, Johor 81310, Malaysia; jddaniel@utm.my

[6] School of Chemical and Energy Engineering, Faculty of Engineering, Universiti Teknologi Malaysia (UTM), Skudai, Johor 81310, Malaysia

[7] City of Scientific Research and Technology Applications (SRTA), New Burg Al Arab, Alexandria 21934, Egypt

\* Correspondence: noshinilyas@yahoo.com (N.I.); henshasy@ibd.utm.my (H.E.E.)

**Abstract:** Halo-tolerant plant growth-promoting rhizobacteria (PGPR) have the inherent potential to cope up with salinity. Thus, they can be used as an effective strategy in enhancing the productivity of saline agro-systems. In this study, a total of 50 isolates were screened from the rhizospheric soil of plants growing in the salt range of Pakistan. Out of these, four isolates were selected based on their salinity tolerance and plant growth promotion characters. These isolates ($SR_1$. $SR_2$, $SR_3$, and $SR_4$) were identified as *Bacillus* sp. (KF719179), *Azospirillum brasilense* (KJ194586), *Azospirillum lipoferum* (KJ434039), and *Pseudomonas stutzeri* (KJ685889) by 16S rDNA gene sequence analysis. In vitro, these strains, in alone and in a consortium, showed better production of compatible solute and phytohormones, including indole acetic acid (IAA), gibberellic acid (GA), cytokinin (CK), and abscisic acid (ABA), in culture conditions under salt stress. When tested for inoculation, the consortium of all four strains showed the best results in terms of improved plant biomass and relative water content. Consortium-inoculated wheat plants showed tolerance by reduced electrolyte leakage and increased production of chlorophyll a, b, and total chlorophyll, and osmolytes, including soluble sugar, proline, amino acids, and antioxidant enzymes (superoxide dismutase, catalase, peroxidase), upon exposure to salinity stress (150 mM NaCl). In conclusion, plant growth-promoting bacteria, isolated from salt-affected regions, have strong potential to mitigate the deleterious effects of salt stress in wheat crop, when inoculated. Therefore, this consortium can be used as potent inoculants for wheat crop under prevailing stress conditions.

**Keywords:** salinity; PGPR; wheat; compatible solutes; antioxidant enzymes

## 1. Introduction

Globally, the production rate of agriculture is far less than the estimated food requirement of the ever-increasing population and the gap will be widened over time [1] (GAP Report, 2018).

Agro-ecosystems are influenced by environmental and climatic conditions, farming techniques, and management practices. It is estimated that internationally, salinity affects 22% of the total cultivated and 33% of the total irrigated agricultural area, which is increasing at an alarming rate of 10% annually. Pakistan is also facing severe salinity issues and a total area of 6.30 million hectares is salt affected, out of which 1.89 million hectares is marked as saline [2].

Due to a higher concentration of sodium chloride (NaCl), plants growing in salt-affected soils suffer from both hyperosmotic and hyperionic effects. These stresses result in reduced water uptake; altered ion and mineral absorption rates; increased production of reactive oxygen species, causing disorganization of the cell membrane; and reduction of metabolic activities [3]. Halophytes adapt themselves to saline conditions by adjusting their physiological activities, maintaining their water balance by osmotic adjustments, producing compatible solutes, and modifying the antioxidant system [4]. Some plants overcome salinity stress through the production of osmolytes, particularly glycine betaine, proline, soluble sugars, and proteins [5].

Improvement in the crop yield of saline soils requires a multidimensional approach consisting of salt-tolerant varieties or amelioration by chemical neutralizers, but there is a dire need for eco-friendly sustainable approaches. Rhizobacteria, showing potential to improve plant growth, are termed as plant growth-promoting rhizobacteria (PGPR) [6]. PGPR have the potential to improve plant growth through various mechanisms, including better plant growth, the production of phytohormones, and amelioration of stresses [7]. Due to the natural coping mechanisms of PGPR, their inoculation can help the amelioration of various abiotic stresses in plants. PGPR inoculation can help to improve the growth and yield of crops, particularly in regions prone to drought and salt stress [8,9]. Natural halotolerant PGPR strains have better potential for the amelioration of salt stress in regional crops for sustainable yields. These native PGPR strains are well acclimated to indigenous conditions and the plant–microbe interactions can help the plants to tolerate stress [10].

In this study, native halotolerant PGPR strains were isolated from local saline soils, and their ability to promote plant growth when inoculated under salt stress was investigated. The objective of the present research was to focus on the evaluation of isolated bacterial strains to stimulate salinity tolerance and the promotion of wheat growth, as well as the identification and characterization of the candidate strain both bio-physiochemically and genetically. This study provides a basis to identify and characterize PGPR from natural saline conditions and testing their potential for improving salinity tolerance in wheat, the major staple crop across the world.

## 2. Materials and Methods

### 2.1. Soil Sampling and Physicochemical Analysis

The rhizospheric soil of four halophytes namely, *Abutilon bidentatum*, *Maytenus royleanus*, *Leptochloa fusca* (Kallar grass), and *Dedonia viscose*, was collected from a salt range of Pakistan (313–360 m.a.s.l; 32°23–33°00 north latitude and 71°30–73°30 east longitude). The rhizospheric soil was sieved and stored at 4 °C for future analysis. Rhizospheric soil was analyzed for pH and electrical conductivity (EC) [11], soil texture, macro and micronutrients [12], and available nutrients [13].

### 2.2. Strain Isolation and HaloTolerance Assay

Rhizobacteria were isolated from rhizospheric soil of *Abutilon bidentatum*, *Maytenus royleanus*, *Leptochloa fusca*, and *Dedonia viscose* by using the serial dilution and spread plate techniques [14]. The soil suspension was made by adding 1 g of soil in 9 mL of Milli-Q distilled water. An aliquot of soil suspension was inoculated on Luria-Bertani (LB) agar plates and incubated at 28 ± 2 °C for 48 h. The obtained colonies were purified by sub-culturing. The colony-forming unit (CFU) was calculated according to the formula given by [15]:

$$CFU/g = (\text{colonies number} \times \text{dilution factor/volume of inoculum}).$$

Distinct bacterial colonies were examined for colony characteristics (shape, size, margin, elevation, appearance, texture, pigmentation, and optical properties) as well as for cellular characteristics (cell shape, gram testing) [16]. QTS-24 kits were used to determine the carbon/nitrogen (C/N) source utilization pattern of bacterial isolates. Isolated bacterial strains were tested for their halotolerance abilities by growing them in LB media supplemented with NaCl (2%, 4%, 6%, 8%, 10%, 15%) [16].

### 2.3. Plant Growth-Promoting (PGP) Traits

All the bacterial isolates were evaluated for their PGP characteristics. Phosphorous (P) solubilization was done by spot inoculating overnight grown cultures onto pikovaskaya's agar (Sigma) containing tri-calcium phosphate as an insoluble P source [17]. The colonies, which produced clearing zones in the pikovaskaya's agar plates, were considered positive for phosphorous solubilization. Total solubilized phosphate was measured by using the phosphomolybdate blue color method [18]. Modified pikovaskaya's broth medium was inoculated with each strain and incubated at 30 °C for 5 days. The cultures were centrifuged at 6000 rpm for 15 min. The supernatant (500 μL) was mixed with 40 μL of 2,4-dinitrophenol, after which 20 μL of dilute sulfuric acid were added, followed by 5 mL of chromogenic reagent, and the volume was diluted to 50 mL using sterilized water and absorbance was recorded at 680 nm. Siderophore production was done by spot inoculation on chrome azurol S (CAS) media as described by Schwyn and Neilands [19]. Bacterial strains were spot inoculated on petri plates containing CAS media. An uninoculated plate was taken as the control. After inoculation, plates were incubated at 28 °C for 5–7 days and observed for the formation of an orange zone around the bacterial colonies. Bacterial isolates were tested for hydrogen cyanide production through the method of Lorck [20]. Bacterial strains were streaked on nutrient agar medium (pre-soaked in 0.5% picric acid and 2% sodium carbonate $w/v$), supplemented with glycine (4.4 g/L). Plates were sealed with parafilm paper and incubated at 30 °C for 4 days. The appearance of an orange or red color indicates the production of hydrogen cyanide.

### 2.4. Germination Experiment

This experiment was carried in the Plant Physiology Laboratory of PMAS-Arid Agriculture University. Seeds of the wheat variety (Galaxy 2013) obtained from the National Agricultural Research Centre, Islamabad were surface sterilized by treatment with sodium hypochlorite (1%) solution for 5 min. After, seeds were successively washed with distilled water. All the isolated strains were tested for germination attributes. Sterilized seeds of wheat were placed in pre-soaked filter paper in Petri dishes. NaCl solution (50 mM, 100 mM, 150 mM, 200 mM) was given instead of normal water. The germination experiment was carried out under laboratory conditions with an average photoperiod of 10 h day/14 h night at 24 °C. The germination percentage, seedling vigor index, and promptness index were measured for each treatment [21]. Four strains were selected for further analysis, based upon their efficacy in the germination experiment and were labeled as $SR_1$, $SR_2$, $SR_3$, and $SR_4$.

### 2.5. Production of Osmolytes

To analyze proline and total soluble sugars, the supernatant of PGPR grown in LB broth supplemented with NaCl concentrations (0%, 2%, 4%, 6%, 8%, and 10%) were analyzed as described by Upadhyay et al. [22]. For the estimation of the proline contents, centrifugation of the culture broth was done at $1000\times g$ for 10 min and the supernatant was used for estimation. Total soluble sugar (TSS) was estimated by mixing 1 mL of supernatant with 4 mL of anthrone reagent, the mixture was later boiled in a water bath for 8 min. After rapid cooling, the optical density was measured at 630 nm, and the amount of TSS was calculated from a standard curve.

### 2.6. Phytohormone Production

The ability of four selected halotolerant strains to produce phytohormones (IAA, GA, CK, ABA) in the culture media was measured by the method of Tien et al. [23]. The extraction of hormones was

done by centrifugation of bacterial cultures at 10,000 rpm for 15 min. For adjustment of the pH (2.8), 1 N HCl was used. In the next step, an equal volume of ethyl acetate was used for hormone extraction. The resulting solution was evaporated at 35 °C and the end residue was mixed in 1500 μL of methanol. Finally, the samples were run on High Performance Liquid Chromatography (HPLC) (Agilent 1100), which had a C18 column (39 × 300 mm) and a UV detector. For standardization of HPLC, pure grade chemicals of the hormones IAA, CK, GA, and ABA (Sigma Chemical Co., St. Louis, MO, USA) were dissolved in HPLC-grade methanol and were used. The wavelength used for the detection was as follows: IAA at 280 nm; and GA, CK, and ABA at 254 nm. The phytohormone content of LB media, without inoculum, was used to normalize the data.

### 2.7. 16S rRNA Gene Sequence and Phylogenetic Analysis

DNA was extracted from pure LB broth cultures as described by Chen and Kuo [24]. Amplification of genomic DNA of isolated strains was done as described by Weisburg et al. [25]. The PCR was carried out for amplification of the 16S rRNA gene with universal nucleotide sequence forward primer (fd1) AGAGTTTGATCCTGGCTCAG, and reverse primer (rd1) (AAGGAGGTGATCCAGCC). DNA was purified and sequenced on an automated sequencer by gel purification kits (JET quick, Gel Extraction Spin Kit, GENOMED). The strains were identified by using a nearly complete sequence of the 16s rRNA gene on (BLAST) NCBI by comparing sequence homology with other strains. The maximum parsimony method was used for the analysis of evolutionary linkages [26].

### 2.8. Plant Inoculation

A pot experiment was conducted in the greenhouse of the Botany Department, PMAS-AAUR, Rawalpindi. A complete randomized design was applied with three replications. Each selected halotolerant strain was grown overnight in LB media. To obtain a cell pellet, the supernatant was discarded after centrifugation at 3000 rpm for 3 min. The cell pellet was washed three times with autoclaved water and the absorbance was recorded with a spectrophotometer at 600 nm to obtain the desired concentration, i.e., $10^7$ CFU. Ten sterilized seeds were sown in each pot (containing 10 kg of soil) in the greenhouse with the day 10 h/14 h night at a temperature of 21/15 °C. Soil moisture was maintained at 15 ± 1%. Four strains and their consortium were evaluated under two treatment controls and 150 mM NaCl stress. The salt level was maintained with EC of 4.0 dS m$^{-1}$ (first irrigation) or 8.5 dS m$^{-1}$ (second irrigation). Plants were harvested after 45 days of sowing. Fresh and dry biomass was recorded. Leaf area was measured with the help of a leaf area meter. All the samples were collected in zipper bags and stored at −20 °C freezer for further biochemical assays. The percent of water content was determined by measuring the ratio between the fresh and dry weight of the upper fully developed leaf by using the following formula [27]:

$$RWC = [FW - DW]/[TW - DW] \times 100 \tag{1}$$

### 2.9. Electrolyte Leakage (%)

Electrolyte leakage was determined by the method of Srairam [28]. Leaf discs weighing 0.1 g were heated in 10 mL of distilled water for 30 min at 40 °C and the electrical conductivity (C1) was recorded. The same discs were then heated at 100 °C and again electrical conductivity (C2) was recorded. Whereas, calculations were done by the following formula:

$$MSI = [1 - (C1/C2)] \times 100 \tag{2}$$

### 2.10. Chlorophyll and Carotenoid Content

Leaf chlorophyll a, b, total chlorophyll, and carotenoid contents were estimated by the method of Arnon [29]. Fresh leaves (0.5 g) were ground in 10 mL of 80% acetone. The readings of the filtrate were measured at 470 nm, 663 nm, and 645 nm. Calculations were done by the following equations:

$$Chla\ (mg/g) = [12.7A_{663} - 2.69A_{645}]\ (v/w) \tag{3}$$

$$Chlb\ (mg/g) = [22.9A_{645} - 4.68A_{663}]\ (v/w) \tag{4}$$

$$Total\ chlorophyll\ (mg/g) = [(20.2A_{645} + 8.02A_{663})\ v/w] \tag{5}$$

$$Carotenoids\ content(mg/g) = (1000\ A_{470} - 1.8\ Chl_a - 85.02\ Chl_b)/198 \tag{6}$$

where A is the optical density at a specific wavelength.

### 2.11. Proline Content

Proline contents were determined by following the protocols of Bates [30]. Fresh leaves (0.5 g) were homogenized with 10 mL of sulfosalicylic acid (3.0%). The solution was filtered, and the filtrate was mixed with equal amounts of glacial acetic acid and ninhydrin reagent. The mixture was heated for 1 h in a water bath at 90 °C and the reaction was stopped by transferring the mixture to ice. Toluene (1 mL) was added to the mixture and the solution was mixed and the solution separated into two layers. The upper layer was isolated in separate test tubes and the reading was measured at 520 nm. Proline was determined as follows:

$$Proline = (Reading\ of\ sample \times Diluted\ concentration \times K\ value)/material\ weight \tag{7}$$

### 2.12. Total Soluble Sugar and Amino Acid

Soluble sugars were estimated after the method of Dubois et al. [31]. Ground plant tissue (0.1 g) was mixed with 3 mL of 80% methanol. The solution was heated in a water bath for 30 min at 70 °C. An equal volume of extract (0.5 mL) and 5% phenol was mixed with concentrated sulphuric acid (1.5 mL) and was again incubated in the dark for 30 min. The absorbance of the sample was checked at 490 nm and the calculations were done by applying the following formula:

$$Sugar\ (\mu g/mL) = Absorbance\ of\ sample \times Dilution\ factor \times K\ value \tag{8}$$

Fresh tissue in grams.

The standard curve was prepared for glucose solution, which was used for the determination of the amount of sugar, expressed in mg $g^{-1}$ fw$^{-1}$.

The Ninhydrin method was used for the determination of free amino acids [32]. Leaf extract (1 mL) was mixed with the same volume of 0.2 M citrate buffer (pH-5) and 80% ethanol, and 2 mL of the ninhydrin reagent. The absorbance of the reaction mixture was taken to 570 nm. Amino acids were computed with the equation:

$$Amino\ acids = Absorption \times volume \times Diluted\ concentration/Sample\ weight \times 1000.$$

The amino acid, leucine, was used for preparing the standard curve, and results were expressed in mg of amino acid per g of dry tissue.

### 2.13. Total Protein Content

The concentration of protein was quantified by the Bradford assay [33]. Bovine serum albumin was used as a standard. Proteins were extracted by dissolving 0.2 g of leaf samples in 4 mL of sodium

phosphate buffer (pH 7), and 0.5 mL of the extract was mixed with 3 mL of Comassive bio red dye. The optical density of the solution was measured at 595 nm. Protein was determined by:

$$\text{Protein} = \text{Reading of extract} \times \text{Diluted concentration} \times \text{value of K/sample weight} \tag{9}$$

### 2.14. Antioxidant Enzyme Assay

Enzyme extract was prepared by grinding one gram of leaf in liquid nitrogen. The obtained powder was added in 10 mL of 50 mM phosphate buffer (pH 7.0) and was mixed with 1 mM Ethylene Diamine Tetra Acetic acid (EDTA) and 1% polyvinylpyrrolidone (PVP). The whole mixture was centrifuged at 13,000× $g$ for 20 min at 4 °C. The supernatant was used for the enzyme assay.

The catalase (CAT) content was estimated by observing the degradation of $H_2O_2$ at 240 nm [34]. Catalase activity (U mg protein$^{-1}$) was calculated from the molar absorption coefficient of 40 mm$^{-1}$cm$^{-1}$for $H_2O_2$. Peroxidase dismutase (POD) was determined by following the procedure of Rao [35]. The reaction mixture consisted of 10 μL of crude enzyme extract, 20 μL of 100 mM guaiacol, 10 μL of 100 mM $H_2O_2$, and 160 μL of 50 mM sodium acetate (pH 5.0). Absorbance was recorded at 450 nm.

Superoxide dismutase (SOD) activity was done by using the procedure of Giannopolitis and Ries [36]. The composition of the reaction mixture was 50 mM sodium phosphate buffer (pH 7.8), 0.1 M tris-HCL, 14 mM methionine, 1.05 mM riboflavin, 0.03% TritonX-100, 50 mM nitroblue tetrazolium chloride (NBT), 100 mM EDTA, and 20 μL enzyme extracts. After adding riboflavin, the glass tubes were illuminated for 5 min, and reactions were stopped by turning off lights. The absorbance was recorded at 560 nm.

### 2.15. Statistical Analysis

Three replicates were used for the mean and standard deviation values of the data. The obtained data were further analyzed by Duncan's multiple range tests using MSTAT-C version 1.4.2. The correlation coefficient of the data was done using the software Statistix version 8.1. Mean values were compared by the least significant difference (LSD) at $p \leq 0.05$ [37]. The heatmap for the correlation coefficient was prepared by using web tool clustvis (https://biit.cs.ut.ee/clustvis/).

## 3. Results

### 3.1. Soil Analysis

Analysis of the rhizospheric soil samples of all four plants showed the soil was sandy clay loam with an EC range of 0.76–0.85 dSm$^{-1}$, pH in the range of 7.99–8.12, high Na/K ratio, and a low concentration of nutrients (Table 1).

**Table 1.** Physiochemical properties of the rhizosphere soil and rhizobacterial population.

| Host Plant Species | pH | EC (dSm$^{-1}$) | Soil Texture | SAR (mmol/L) | OC (%) | Macronutrient (meq/L) | | | | | | Available Nutrients (kg/ha) | | | PGPR Population (cfu $\times$ 10$^5$ g$^{-1}$ of Soil) |
|---|---|---|---|---|---|---|---|---|---|---|---|---|---|---|---|
| | | | | | | CO$_3$ | HCO$_3$ | Cl | Ca$^+$ | Na$^+$ | K$^+$ | N | P | K | |
| *Abutilom bidiantum* | 7.99 | 0.82 | Sandy clay loam | 39.6 | 0.72 | 3.9 | 15 | 50 | 3 | 82.95 | 0.3 | 240 | 200 | 320 | 69 |
| *Maytenus royleanus* | 8.12 | 0.85 | Sandy clay loam | 40.3 | 0.54 | 4 | 14 | 48 | 2.5 | 80.6 | 0.4 | 238 | 196 | 325 | 64 |
| Kallar grass | 7.80 | 0.76 | Sandy clay loam | 42.5 | 0.62 | 4.2 | 17 | 47 | 2.8 | 78.9 | 0.5 | 243 | 190 | 315 | 62 |
| *Dedonia viscoca* | 8.01 | 0.79 | Sandy clay loam | 38.1 | 0.64 | 4.5 | 17.5 | 48 | 3.1 | 81.2 | 0.4 | 237 | 205 | 330 | 65 |

### 3.2. Isolation and Screening of Salt-Tolerant PGPR Strains

A total of 50 isolates were obtained from the rhizospheric soil of four halophytic plants. Among all isolates, 90% of colonies were round, creamy, and had entire margins (Supplementary Materials Table S1). Further, 78% of isolates were Gram-negative and rod-shaped (Supplementary Materials Table S2).

In the halotolerant assay, 70% of strains were able to grow up to 6%, 20% strains showed tolerance at 10%, while four strains $SR_1$, $SR_2$, $SR_3$, and $SR_4$ were able to grow at 15% NaCl (Supplementary Materials Table S2). These four strains also showed positive results for phosphorous solubilization, hydrogen cyanide, and siderophore production (Supplementary Materials Table S3).

### 3.3. Effect of Bacterial Isolates on Germination of Wheat

Salt stress resulted in a considerable reduction in the germination parameters of the wheat seeds. Under salt-stressed conditions, the seedling vigor index and germination index showed a 12.5% and 31% decrease compared to the control. Though most of the strains showed a significant increase in seed germination, four strains $SR_1$, $SR_2$, $SR_3$, and $SR_4$ showed prominent results (14.28%, 35%, 42%, and 55%), respectively, as compared to the non-inoculated control under the salt stress condition (Supplementary Materials Table S4).

### 3.4. Identification of Isolates

Initially, the four strains were identified based on the C/N source utilization pattern (Supplementary Materials Table S5). Molecular identification of the screened halotolerant strains was done based on 16S rRNA sequences and on the comparison of the 1500-bp sequence of 16S rRNA gene subjected to BLAST to confirm the relatedness with other bacterial strains. The isolate $SR_1$ (1485 base pair) was closely related (98% nucleotide identity) to sequences of bacteria annotated as *Bacillus* strain JQ 926435 in the GenBank database. The sequence of $SR_2$ (1480 base pairs) was 99% identical to *Azospirillum brasilense* DQ 288686.1, $SR_3$ (1482 base pairs), and 96% identical to strain *Azospirillum lipoferum* accession no. M. 5906.1. Furthermore, the isolated strain $SR_4$ showed a 99% homology with *Pseudomonas stutzeri* JQ 926435. The accession numbers of the identified strains were obtained from NCBI and are given in Table 2.

**Table 2.** Molecular identification of the isolates based on partial 16S rDNA analysis.

| No | Isolates | Base Pair Length | Similarity (%) | Strain Identification | Accession No. |
|----|----------|------------------|----------------|-----------------------|---------------|
| 1 | $SR_1$ | 1485 | 98% | *Bacillus* sp. | KF719179 |
| 2 | $SR_2$ | 1480 | 99% | *Azospirillum brasilense* | KJ194586 |
| 3 | $SR_3$ | 1482 | 96% | *Azospirillum lipoferum* | KJ434039 |
| 4 | $SR_4$ | 1263 | 99% | *Pseudomonas stutzeri* | KJ685889 |

Further phylogenetic analysis of the identified bacteria was conducted in MEGA4 software to determine their affiliation [38]. The evolutionary history was inferred using the maximum parsimony method [26]. The results are shown in Supplementary Materials Figures S1–S4.

### 3.5. Production of Phytohormones

Based on the halotolerance assays, PGP traits, and germination assay results, four isolates were selected for further analysis. All the halotolerant PGPR strains showed the production of phytohormones in liquid culture (Figure 1). Halotolerant PGPR strains were able to produce IAA (0.5–2.1 µg mL$^{-1}$), gibberellic acid (1.5–2.5 µg mL$^{-1}$), CK (0.39–0.64 µg mL$^{-1}$), and ABA (1.9–3.4 µg mL$^{-1}$). The PGPR strains $SR_2$ and $SR_3$ produced higher concentrations of phytohormones than those of $SR_1$ and $SR_4$; however, the bacterial consortium produced maximum concentrations of IAA (2.1 µg mL$^{-1}$), gibberellic acid (2.5 µg mL$^{-1}$), CK (0.64 µg mL$^{-1}$), and ABA (3.4 µg mL$^{-1}$).

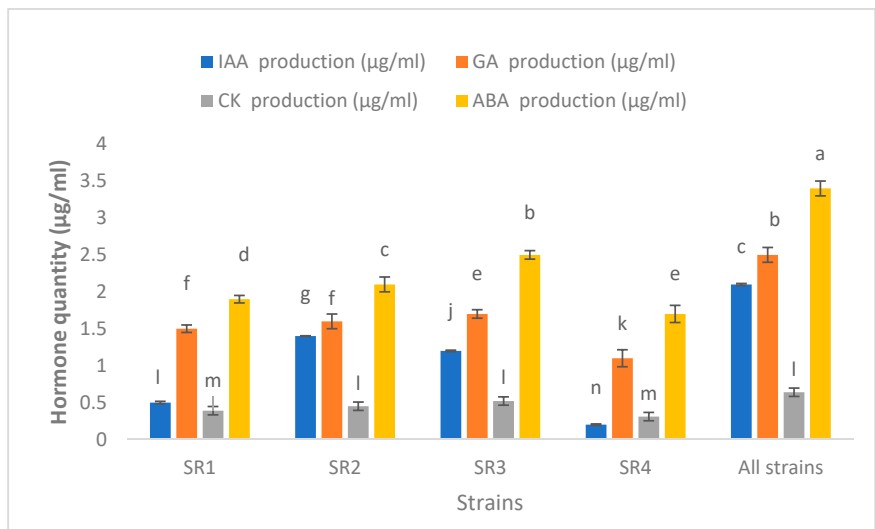

**Figure 1.** Production of phytohormones (Indole Acetic Acid (IAA), Gibberellic Acid (GA), Cytokinin (CK), and Abscisic Acid (ABA) by PGPR strains and their consortium in culture media. (SR$_1$: Inocualted with *Bacillus* sp; SR$_2$: Inocualted with *Azospirillum brasilense*; SR$_3$: Inocualted with *Azospirillum lipoferum*; SR$_4$: Inocualted with *Pseudomonas stutzeri*; Consortium is a combination of all four strains *Bacillus* sp, *Azospirillum brasilense, Azospirillum lipoferum, Pseudomonas stutzeri*). This data displays the means and standard deviation (*n* = 3). Different letters show significant differences between treatments (*p* < 0.05).

### 3.6. Production of Compatible Solutes

A considerable amount of proline was produced by all the screened halotolerant strains when subjected to different salinity levels. Production of proline by SR$_2$ and SR$_3$ was the highest in the 10% saline condition than the control. The maximum amount of proline (12.1 µg mg$^{-1}$) was produced by the bacterial consortium, which was 23% greater than SR$_2$ and SR$_3$. For the carbohydrate contents, a significant amount of soluble sugars was recorded by all the strains (Figure 2). The production of soluble sugars was more pronounced at different salinity levels than the control. The bacterial strains SR$_2$ and SR$_3$ produced a greater amount of (89–111 µg mg$^{-1}$) soluble sugar as compared to the control, but the consortium of bacterial isolates recorded the maximum values at 10% NaCl (222 µg mg$^{-1}$) (Figure 3).

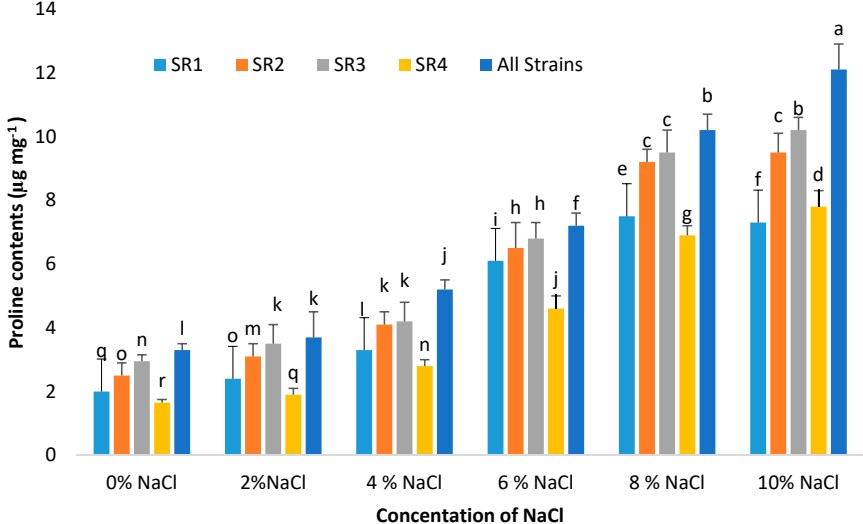

**Figure 2.** Production of proline by PGPR strains and their consortium in culture media supplemented with different concentrations of NaCl (2%, 4%, 6%, 8%, and 10%). The treatment details are the same as in Table 3. This data displays the means and standard deviation (*n* = 3). Different letters show significant differences between treatments (*p* < 0.05).

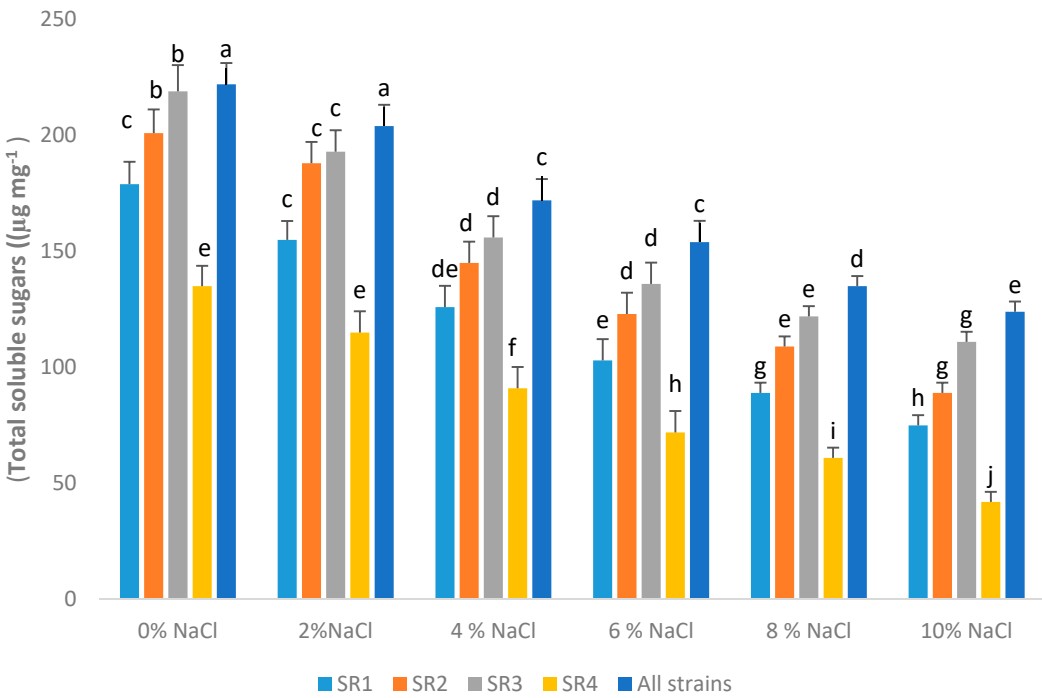

**Figure 3.** Production of total soluble sugar by PGPR strains and their consortium in culture media supplemented with different concentrations of NaCl (2%, 4%, 6%, 8%, and 10%). The treatment details are the same as in Table 3. This data displays the means and standard deviation (*n* = 3). Different letters show significant differences between treatments (*p* < 0.05).

## 3.7. Effect of PGPR Inoculation on the Biomass of Wheat (Triticum aestivum L.) Plants Grown under Salinity Stress

The overall decrease of 30% in the plant biomass of wheat plants was observed due to salt stress. However, the bacterial isolates exerted a significant positive influence on wheat growth and resulted in an increase in the biomass of plants in the control and stressed conditions, respectively. The relative increase in the fresh and dry biomass due to bacterial isolates ranged between 39% and 67% as compared to the uninoculated plants under saline conditions.

The best results were obtained when plants were inoculated with a consortium of all four isolated strains, which caused an increase of 93% in stress and 60% in controlled conditions. Moreover, pronounced results were also encountered for dry biomass, when plants were inoculated with a consortium, which resulted in an increase of 65.4% in salt stress and 78.7% in control conditions (Table 3).

**Table 3.** Effect of inoculation of halotolerant PGPR on the fresh and dry biomass and leaf area of wheat plants grown under salinity stress.

| Treatments | Fresh Biomass (g) | | Dry Biomass (g) | | Leaf Area (cm$^2$) | |
|---|---|---|---|---|---|---|
| | 0 mM | 150 mM | 0 mM | 150 mM | 0 mM | 150 mM |
| Control | 10 ± 1g | 7.3 ± 0.4i | 3.3 ± 0.1c | 2.2 ± 0.04d | 140 ± 12e | 120 ± 17f |
| SR$_1$ | 11 ± 0.5f | 8.2 ± 0.9h | 3.5 ± 0.4c | 2.8 ± 0.06d | 150 ± 15d | 130 ± 14f |
| SR$_2$ | 13.2 ± 0.9e | 10.3 ± 0.7g | 4.1 ± 0.1b | 3.5 ± 0.09c | 167 ± 12c | 140 ± 24e |
| SR$_3$ | 16.9 ± 1.2 b | 13.9 ± 1.4d | 4.7 ± 0.5b | 3.8 ± 0.03 c | 177 ± 17b | 147 ± 17e |
| SR$_4$ | 11.50.98f | 7.8 ± 0.54i | 3.6 ± 0.5c | 3.0 ± 0.08d | 160 ± 14c | 135 ± 12f |
| Consortium | 20.3 ± 1.8a | 14.1 ± 1.9c | 5.9 ± 0.2a | 4.3 ± 0.03b | 186 ± 19a | 152 ± 13d |

This data displays the means and standard deviation (*n* = 3). Different letters show significant differences (*p* < 0.05). (SR$_1$: Inocualted with *Bacillus* sp; SR$_2$: Inocualted with *Azospirillum brasilense*; SR$_3$: Inocualted with *Azospirillum lipoferum*; SR$_4$: Inocualted with *Pseudomonas stutzeri*; Consortium is a combination of all four strains *Bacillus* sp, *Azospirillum brasilense*, *Azospirillum lipoferum*, *Pseudomonas stutzeri*).

### 3.8. Effect on the Membrane Stability Index and Water Content

Results of the percent electrolytic leakage showed that the inoculation remains significant under stress as well as normal conditions However, co-inoculation with bacterial consortium successfully decreased (34%) the ionic discharge at the 150 mM NaCl level compared to the control (Table 4). Furthermore, the percent of water content showed a significant reduction of 33% in wheat plants under salt stress as compared to the uninoculated control plants. More pronounced results were obtained with $SR_2$ and $SR_3$, causing an increase of 10.5% and 17.54% in the stress condition. The consortium-inoculated plants recorded the maximum amount of water of 21% and 17.64% in the stress and control conditions. A similar trend was observed by $SR_1$ and $SR_4$ (Table 4).

**Table 4.** Effect of inoculation of halotolerant PGPR strains on the leaf water content and electrolyte leakage of wheat plants grown under salinity stress.

| Treatments | Percent Water Content | | Electrolyte Leakage (%) | |
|---|---|---|---|---|
| | 0 mM | 150 mM | 0 mM | 150 mM |
| Control | 85 ± 1.5b | 57 ± 0.9e | 33 ± 0.4d | 55 ± 0.5a |
| SR$_1$ | 86.3 ± 1.6b | 60 ± 1d | 30 ± 0.3d | 50 ± 0.45a |
| SR$_2$ | 89 ± 1.9b | 63 ± 1.4d | 26.2 ± 0.25e | 41.3 ± 0.33c |
| SR$_3$ | 95 ± 2.1a | 67 ± 1.7d | 25.7 ± 0.4e | 42.08 ± 0.11c |
| SR$_4$ | 88.7 ± 1.9b | 61 ± 1.15d | 31.2 ± 0.22e | 47.8 ± 0.44b |
| Consortium | 97 ± 2.0a | 70 ± 1.75c | 22.1 ± 0.22f | 35.2 ± 0.23d |

This data displays the means and standard deviation ($n = 3$). Different letters show significant differences ($p < 0.05$). Treatment details are the same as in Table 3.

### 3.9. Chlorophyll Contents

Salinity stress negatively affected the photosynthetic pigments of wheat plants. A considerable decrease of 30.4%, 22%, and 25% was observed in chlorophyll a, b, and total chlorophyll. The response to the consortium was effective ($p \leq 0.05$) and resulted in a 13.23%, 12.49%, 12.9%, and 11.76% increase as compared to the control under salt-stress conditions (Table 5).

**Table 5.** Effect of halotolerant PGPR on the chlorophyll a, chlorophyll b, total chlorophyll, and carotenoid contents of wheat plants grown under salinity stress.

| | Chlorophyll a (mg/g Fresh Weight) | | Chlorophyll b (mg/g Fresh Weight) | | Total Chlorophyll (mg/g Fresh Weight) | | Carotenoid (mg/g Fresh Weight) | |
|---|---|---|---|---|---|---|---|---|
| Treatments | 0 mM | 150 mM | 0 mM | 150 mM | 0 mM | 150 mM | 0 mM | 150 mM |
| Control | 1.06 ± 0.01d | 0.59 ± 0.01h | 0.27 ± 0.02d | 0.12 ± 0.01h | 1.18 ± 0.10e | 0.86 ± 0.05f | 46.9 ± 0.1f | 65.8 ± 0.15k |
| SR$_1$ | 1.13 ± 0.03b | 0.75 ± 0.03g | 0.29 ± 0.04b | 0.13 ± 0.02g | 1.26 ± 0.09d | 1.01 ± 0.03k | 47.3 ± 0.3e | 67.6 ± 0.5i |
| SR$_2$ | 1.18 ± 0.04c | 0.81 ± 0.02f | 0.32 ± 0.03c | 0.15 ± 0.02f | 1.33 ± 0.7c | 1.13 ± 0.02j | 50.5 ± 0.4c | 69.8 ± 0.4h |
| SR$_3$ | 1.2 ± 0.05c | 0.85 ± 0.04g | 0.33 ± 0.05c | 0.17 ± 0.03f | 1.37 ± 0.8b | 1.18 ± 0.04k | 51.1 ± 0.2b | 69.2 ± 0.5i |
| SR$_4$ | 1.12 ± 0.02b | 0.77 ± 0.03b | 0.28 ± 0.01b | 0.13 ± 0.02g | 1.13 ± 0.6f | 1.05 ± 0.01d | 48.2 ± 0.4d | 68.1 ± 0.6j |
| Consortium | 1.4 ± 0.04a | 0.9 ± 0.02e | 0.35 ± 0.05a | 0.19 ± 0.04e | 1.59 ± 0.5a | 1.25 ± 0.03h | 52.8 ± 0.6a | 70.4 ± 0.8g |

This data displays the means and standard deviation ($n = 3$). Different letters show significant differences ($p < 0.05$). Treatment details are the same as in Table 3.

### 3.10. Proline Contents

Salinity stress increased proline accumulation in wheat plants. A considerable increase of 50% in the proline content of wheat plants was recorded in saline stress conditions as compared to their respective control. Inoculation with halotolerant PGPR increased the levels of proline in the leaves. All four inoculants increased the proline contents in the range of 18–36%, respectively. The accumulation of proline was maximum in consortium-treated plants, with an increase of 46.67% under stress conditions (Table 6).

**Table 6.** Effects of halotolerant PGPR on the total soluble sugar, amino acid, protein, and proline contents of wheat plants grown under salinity stress.

| Treatments | Total Soluble Sugar ($\mu$g g$^{-1}$ FW) | | Total Amino Acid ($\mu$g g$^{-1}$ FW) | | Proline ($\mu$g g$^{-1}$ FW) | |
| --- | --- | --- | --- | --- | --- | --- |
| | 0 mM | 150 mM | 0 mM | 150 mM | 0 mM | 150 mM |
| Control | 27 ± 2d | 33 ± 5i | 330 ± 10g | 368 ± 20e | 40 ± 03d | 120 ± 5j |
| SR$_1$ | 29 ± 3d | 35 ± 7h | 345 ± 12 f | 379 ± 17e | 44 ± 05d | 128 ± 6i |
| SR$_2$ | 31 ± 5c | 39 ± 6g | 360 ± 15e | 401 ± 27c | 51 ± 3c | 130 ± 7g |
| SR$_3$ | 33 ± 3c | 33 ± 1.0f | 370 ± 24e | 420 ± 25b | 54 ± 3b | 135 ± 9h |
| SR$_4$ | 29 ± 4 d | 36 ± 8h | 350 ± 12f | 387 ± 24.4d | 43 + 3 c | 125 ± 5i |
| Consortium | 39 ± 5 c | 43 ± 5e | 381 ± 10d | 439 ± 15a | 57 ± 4b | 145 ± 7f |

This data displays the means and standard deviation (*n* = 3). Different letters show significant differences ($p < 0.05$). Treatment details are the same as in Table 3.

### 3.11. Amino Acid Content

The amino acid content was highest in the consortium of halotolerant PGPR strains, with an increase of 19.29% and 15.54% under salt stress and control conditions. Moreover, plants inoculated with SR$_2$ and SR$_3$ contained 10% and 14.1% greater concentrations of amino acids as compared to the uninoculated stressed plants (Table 6).

### 3.12. Total Soluble Sugar

Salinity stress produced a significant increase of 12.5% for the soluble sugar contents of wheat plants as compared to the control. The best outcomes were obtained when plants were inoculated with SR$_2$ and SR$_3$, which resulted in an increase of 9.52% and 15.87%, respectively, under stress conditions. However, a more prominent effect was revealed with the inoculation of a consortium of strains, with an increase of 28.57% and 23.2%, respectively, under the stress and control condition (Table 6).

### 3.13. Antioxidants Enzyme Assay

The antioxidant enzymes of the wheat plants showed a significant increase under salinity stress. Inoculation with all four halotolerant PGPR improved the production of antioxidant enzymes in plants. However, the best results were shown by the consortium of all strains. The consortium increased the superoxide dismutase activity by 21.4% as compared to stressed plants. Similarly, a significant increase of 16% in the catalase activity was recorded by the inoculation with the consortium. A significant increase of 34.4% in the peroxidase content of plants was recorded as compared to the control (Table 7).

**Table 7.** Effects of halotolerant PGPR on the antioxidant enzymes activity of wheat plants grown under salinity stress.

| Treatments | Superoxide Dismutase (EU mg$^{-1}$ Protein) | | Catalase (EU mg$^{-1}$ Protein) | | Peroxidase (EU mg$^{-1}$ Protein) | |
| --- | --- | --- | --- | --- | --- | --- |
| | 0 mM | 150 mM | 0 mM | 150 mM | 0 mM | 150 mM |
| Control | 0.74 ± 0.06k | 1.83 ± 0.02f | 2.5 ± 0.03h | 4.13 ± 0.02f | 144 ± 3f | 255 ± 5f |
| SR$_1$ | 0.76 ± 0.04j | 1.85 ± 0.01e | 2.7 ± 0.02k | 4.3 ± 0.04e | 148 ± 7j | 260 ± 4d |
| SR$_2$ | 0.8 ± 0.03i | 1.9 ± 0.04c | 3.01 ± 0.04i | 4.7 ± 0.09c | 153 ± 4i | 263 ± 6c |
| SR$_3$ | 0.82 ± 0.05h | 1.91 ± 0.03b | 3.12 ± 0.05h | 4.8 ± 0.10b | 155 ± 6h | 267 ± 7b |
| SR$_4$ | 0.78 ± 0.3j | 1.85 ± 0.4d | 2.6 ± 0.04j | 4.5 ± 0.05d | 150 ± 4j | 257 ± 5d |
| Consortium | 0.86 ± 0.07g | 1.96 ± 0.05a | 3.25 ± 0.05g | 5.05 ± 0.04a | 162 ± 3g | 270 ± 6a |

This data displays the means and standard deviation (*n* = 3). Different letters show significant differences ($p < 0.05$). Treatment details are the same as in Table 3.

### 3.14. Heatmap Responses of Pearson's Correlation Coefficient (r)

From the heat map analysis, the data of the osmolyte production, electrolyte leakage, chlorophyll contents, antioxidant enzymes, and halotolerant PGPR showed positive correlations (Figure 4). A comparative analysis of the parameters related to salinity tolerance (presented by green boxes)

showed that salinity tolerance had a positive correlation with amino acid, osmotic potential, soluble sugars, proline, SOD, POD, and CAT activities (Figure 5).

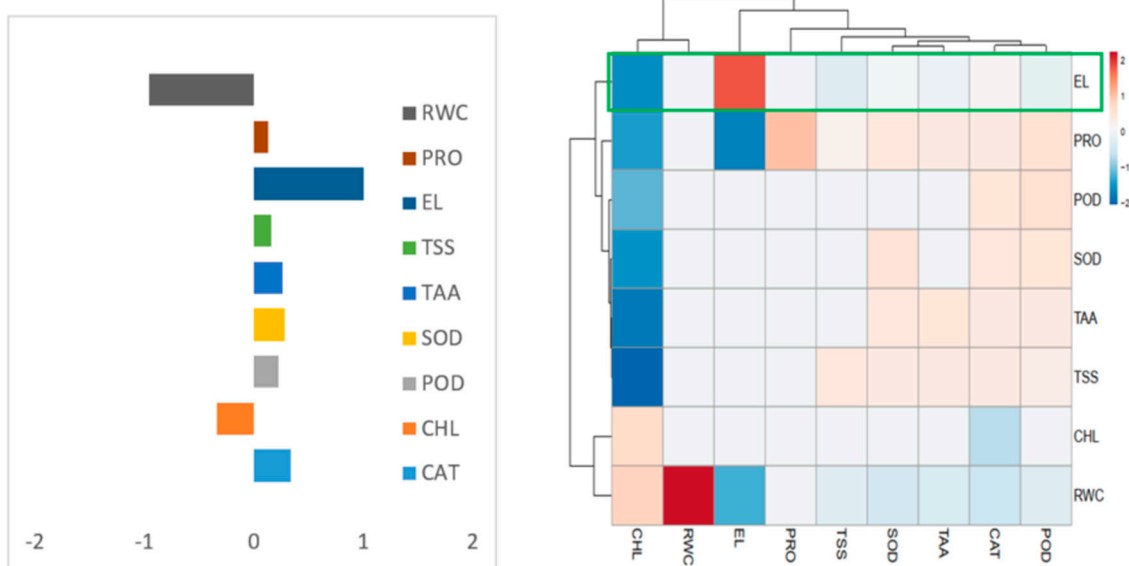

**Figure 4.** Heatmap of the correlation coefficient (r) for the antioxidant enzymes, stress determinants, and relative water content of wheat leaves treated with bacterial isolates and their consortium. Whereas, EL = Electrolyte leakage, Pro = Proline, POD = Peroxidase, SOD = Superoxide dismutase, CHL = Total chlorophyll, TAA = Total amino acids, TSS = Total soluble sugars, RWC = relative water content.

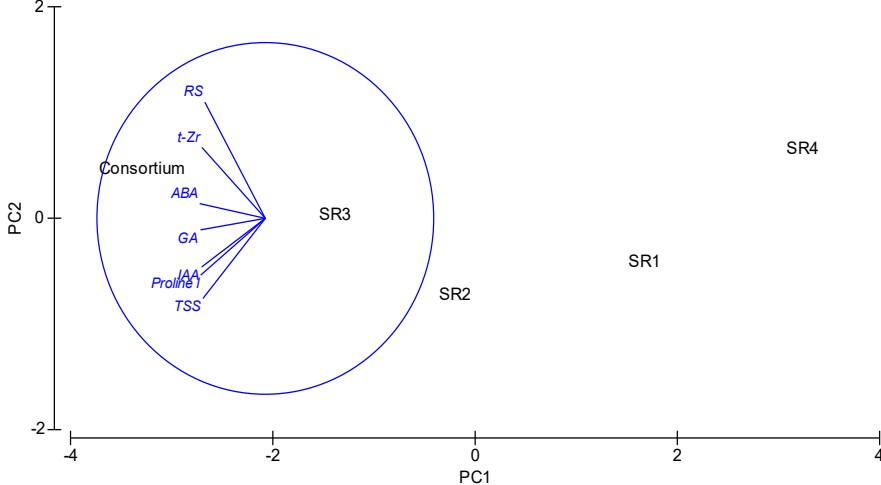

**Figure 5.** Principle component analysis (PCA) of phytohormones, proline, total soluble sugars, and reducing sugars of halotolerant bacterial isolates and their consortium grown under salt stress in culture conditions. Whereas, IAA = Indole acetic acid, GA = gibberellic acid, CK = Trans zeatin riboside, RS = Reducing sugars, TSS = Total soluble sugars.

## 4. Discussion

Soil bacteria associated with rhizosphere have been known as growth promotors as well as biotic and abiotic stress alleviators [8]. Bacteria associated with the roots of halophytes and saline soil, capable of tolerating higher levels of salts, are termed as halotolerant [39]. In the current study, bacterial isolates $SR_1$, $SR_2$, $SR_3$, and $SR_4$ showed the best salt tolerance abilities among all 50 bacterial isolates from the roots–soil interface of plants growing in the saline area. Phenotypic and molecular genotyping (16S RNA sequencing) of four potent isolates proved that $SR_2$ and $SR_3$ strains belong to

the *Azospirillum* genus (*Azospirillum brasilense* and *Azospirillum lipoferum*) and the other two ($SR_I$ and $SR_4$) belong to the genus *Bacillus* (*Bacillus sturtezi*) and *Pseudomonas* (*Paeudomonas stutzeri*) (Table 2). These beneficial PGPR belonged to different genera, which indicate that plant growth promotion has been distributed across different taxons Halotolerant strains from the genera of *Pseudomonas*, *Bacillus*, *Azospirillum*, *Klebsiella*, and *Ochromobacter* have shown remarkable performance in the amelioration of salt stress in a wide range of crops [40].

Halotolerant PGPR has been reported to promote plant growth as well as mitigate salinity stress [41]. In the current study, we attempted to identify the key mechanisms used by halotolerant strains to alleviate the salinity stress in wheat plants by regulating plant defense mechanisms. The ability of halotolerant PGPR to produce phytohormones is associated with improved growth of plants under saline conditions [42]. The halotolerant PGPR produced IAA, GA, CK, and ABA. The results showed that *Azospirillum* strains produced higher amounts of GA, IAA, and CK than those of *Bacillus* and *Pseudomonas* strains in liquid media (Figure 1). The production of hormones by halotolerant PGPR is thoroughly supported by previous literature and many halotolerant strains of *Azotobacter*, *Bacillus*, *Arthrobacter*, *Azospirillum*, and *Pseudomonas* have been shown to produce IAA, GA, CK, and ABA [43]. These phytohormones regulate the stress defense responses in plants. They influence all aspects of plant growth, like cell wall elongation (IAA), cell division (CK), germination (gibberellin), and stress tolerance (ABA) [44–46]. Various reports suggest that these phytohormones produced under salinity stress help plants to survive and impart tolerance in them under abiotic stresses [46].

Here, the results proved that rhizobacteria secrete more compatible solutes (soluble sugars and proline) in culture media supplemented with a higher NaCl (10%) content. Various studies documented that bacterial cells can accumulate a considerable amount of compatible solutes inside their cells, acting as osmolytes and helping them to survive under severe osmotic stress [47].

Salinity is one of the common factors that can limit agricultural productivity due to its effects on seed germination, plant growth, and crop yield. Wheat is an important staple crop, but as it is a moderately salt-tolerant crop, high salt stress strictly limits its growth and development. Salt stress ultimately reduces the crop yield and nutritive value of wheat. The regulation of physiological, enzymatic, and biochemical changes in plants after inoculation with PGPR helps to alleviate salt or drought stress [40,48].

We demonstrated that salinity reduced the growth and development and relative water content of wheat plants. It also caused curling and wilting of leaves, early leaf senescence, and ultimately a reduction in the growth of plants. This is consistent with what was found in a previous study that salinity restricts cell differentiation and the cell cycle due to osmotic and ionic stress, deficiency of nutrients, oxidative damage, and limited water uptake, which affects plant germination, growth development, and physiological processes, ultimately leading to growth inhibition [49].

In this study, a consortium of four strains produced a prominent result for the dry biomass and leaf area than the control and individual inoculants. These results are in line with Walker et al. [50], who reported that inoculation with a consortium of *Azospirillum-Pseudomonas-Glomus* improved the root architecture in maize under salinity. A better adaptability of PGPR to stress conditions is correlated with efficient root colonization, phosphate solubilization, and nitrogen fixation abilities [51]. From the results, it is clear that salinized plants inoculated with halotolerant strains and their consortium exhibited a higher relative water content of leaves. Rakshapal et al. [52] also observed that PGPR-treated plants not only cope with stress but also that these microbes help to maintain higher water levels in comparison to control plants.

Salinity decreases the photosynthetic efficiency of plants and results in the production of reactive oxygen species (ROS), which cause damage to DNA, proteins, and membranes [53]. We described the results of photosynthetic pigments of wheat plants, which showed that treatment with a consortium showed a pronounced effect of reducing the damage caused by salinity on the photosynthetic apparatus. A similar pattern of results was reported by El-Esawi et al. [54], who observed an increase in the photosynthetic efficiency of plants by PGPR inoculation under salinity.

Salt stress can develop more discharge of electrolytes through the misplacement of Ca associated with membranes. As a result, the permeability of the membrane is destroyed and accumulates a higher efflux of electrolytes inside plant cells/tissue [55]. In the current study, the successive increase in the electrolyte leakage of wheat plants was observed at 150 mM salt stress than the control. These results are inconsistent with the Bojórquez-Quintal et al. [56], who found salt stress enhances electrolyte leakage and the generation of reactive oxygen species (ROS), having a detrimental effect on plant growth. Our results showed that inoculation with halotolerant PGPR tends to decrease the injurious effect of saline stress and decrease the potential electrolytic leakage of ions in stress-treated plants. This is consistent with what was found in previous studies [57,58].

In the present study, the concentration of compatible solutes was also increased in inoculated wheat plants under salt stress (Table 7). The accumulation of compatible solutes, particularly proline, free amino acid, and soluble sugar, is correlated with the adaptability of the plant to stress conditions. We reported that halotolerant PGPR produces compatible osmolytes, which help the plants to maintain their ionic balance. PGPR also induce osmolyte accumulation [59] and phytohormone signaling [40], which facilitates plants in overcoming the initial osmotic shock after salinization. In a previous study, it was found that rice inoculation with salt-tolerant *Bacillus amyloliquefaciens* under salinity increased the plant's salt tolerance and affected the expression of genes involved in osmotic and ionic stress response mechanisms [60].

Proline is the most important osmolyte, which is produced in plants by the hydrolysis of proteins under osmotic stress [61]. From the results, it is clear that a consortium of halotolerant PGPR plants improved proline levels under salt stress. These results are in line with Wang et al. [62]. The production of osmolytes helps the plant to maintain a high turgor potential, prevent oxidative damage by scavenging reactive oxygen species, and protect the membrane structure [63].

We also reported a pronounced increase in the production of soluble sugars with a consortium of halotolerant strains in wheat under salinity stress. PGPR can stimulate carbohydrate metabolism and transport, which results in changes in the source–sink relations, photosynthesis, and growth rate. In previous reports, seeds inoculated with *B. aquimaris* strains showed an increased production of total soluble sugars in wheat under salinity conditions, which resulted in higher biomass and plant growth [64].

An increase in the antioxidant enzyme activity of wheat plants grown under salinity stress was observed by a consortium of halotolerant PGPR strains. This indicates that these bacteria can help the plant to combat the deleterious effects of ROS generated during salinity stress. These results tie well with the previous studies, where an increase in antioxidant enzyme activity under salinity stress was proven to be associated with salt tolerance [65]. Moreover, Wang et al. [66] reported that the application of PGPR strains alleviates the oxidative damage induced by abiotic stresses, including salinity, by augmenting the activity of antioxidant enzymes.

## 5. Conclusions

In summary, crop inoculations with halotolerant PGPR consortium can serve as a potential tool for alleviating salinity stress. Halotolerant PGPR strains have developed several mechanisms to cope with salinity, particularly the potential to produce phytohormones and compatible solutes. Halotolerant PGPR strains can induce salinity tolerance in plants by activating key defense mechanisms like the production of osmoregulators as well as activating ROS scavenging enzymes. Natural microflora adapted to saline conditions can be used for the development of microbial consortia for crop inoculation, ultimately leading to the formulation of biofertilizer for salt-stressed areas. However, further investigation is needed to observe their performance in field conditions.

**Supplementary Materials:** The following are available online at http://www.mdpi.com/2073-4395/10/7/989/s1, Table S1: Morphology of isolates from rhizosphere of plants from saline soil, Table S2: Preliminary screening data of isolated strains (+ indicates groeth, − indicates no growth), Table S3: Growth characters of isolated strains, Table S4: Effect of Isolates on germination attributes of wheat, Table S2: Carbon/Nitrogen source utilization pattern

determined by QTS -24 kits, Figure S1: Phylogenetic analysis of strain SR1, Figure S2: Phylogenetic analysis of strain SR2, Figure S3: Phylogenetic analysis of strain SR3, Figure S4: Phylogenetic analysis of strain SR4.

**Author Contributions:** Conceptualization, N.I. Writing—Original Draft, N.I. and R.M. Formal Analysis, H.Y. and W.K. Investigation, R.M. Proofreading, H.E.E. and D.J.D. Editing, H.Y., W.K. and S.I. Formatting, D.J.D. Writing—Review, H.Y., S.I. and W.K. Supervision, N.I. Facilitation, H.E.E. Review, H.E.E. and D.J.D. All authors have read and agreed to the published version of the manuscript.

**Funding:** This research was funded in part by Allcosmos Industries Sdn. Bhd. through research project No. R.J130000.7344.4B200. The APC was supported by UTM-TNCPI research fund.

**Conflicts of Interest:** Authors have no conflict of interest.

**Ethical Statement:** Not applicable.

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
