# Peer review of "Rhizobacteria Isolated from Saline Soil Induce Systemic Tolerance in Wheat (Triticum aestivum L.) against Salinity Stress"

_agronomy, doi:10.3390/agronomy10070989_

Round 1

Reviewer 1 Report

First, I'm not sure how the production of various hormones or other materials in vitro relate to transfer to plants. However, the authors also provided information on plant growth and production of various factors associated with osmotolerance in plants, and this mitigates concerns about the value of in vitro data. In general, the work is complete and well done. It adds to the literature concerning positive effects of PGPR. Importantly, it demonstrates that the beneficial effects of PGPR are independent of genus or species, and, in fact, are distributed across different taxons. I think this would be worth pointing out in the discussion.

A few specific comments:

Throughout, please say either halo-tolerant of halotolerant, rather than halo tolerant. When I first read the Ms, I wondered why PGPR would be resistant to haloes. 

In 3.2, There needs to be a comma after "isolates".

In the last paragraph of the discussion, omit the plural form of osmolyte and add an "s" to "cause".

Author Response

ANNOTATED RESPONSE TO REVIEWER 1

No.

COMMENT

ACTION/JUSTIFICATION

Page No. /Line No.

1.

First, I'm not sure how the production of various hormones or other materials in vitro relate to transfer to plants. However, the authors also provided information on plant growth and production of various factors associated with osmotolerance in plants, and this mitigates concerns about the value of in vitro data. In general, the work is complete and well done. It adds to the literature concerning positive effects of PGPR.

Importantly, it demonstrates that the beneficial effects of PGPR are independent of genus or species, and, in fact, are distributed across different taxons. I think this would be worth pointing out in the discussion

Authors are very grateful to the reviewer for his positive response.

This aspect has been mentioned in discussion.

Page 14; line 353-354

2.

Throughout, please say either halo-tolerant of halotolerant, rather than halo tolerant. When I first read the Ms, I wondered why PGPR would be resistant to haloes. 

The whole manuscript has been revised keenly and suggested correction has been made.

Page 2; line 19

Page 3; line 62

Page 4; line 66

Page 6; line 124

Page 7; line 148

Page 10; line 241

Page 11; line 254

Page 11; line 268

Page 11; line 269

Page 12; line 275

Page 13; line 313

Page 13; line 318

Page 14; line 331

Page 14; line 340

Page 14; line 347

Page 14; line 354

Page 15; line 357

Page 15; line 356

Page 15; line 358

Page 15; line 360

Page 15; line 361

Page 15; line 363

Page 15; line 364

Page 16; line 393

Page 16; line 409

Page 17; line 416

Page 17; line 423

Page 17; line 428

Page 17; line 434

Page 18; line 441

Page 18; line 442

Page 18; line 444

Page 26; line 660

Page 27; line 677

Page 28; line 691

Page 29; line 697

Page 30; line 714

Page 35; line 770

Reviewer 2 Report

The article entitled “Rhizobacteria isolated from saline soil induce systemic tolerance in wheat (Triticum aestivum L.) against salinity stress”, proposed for publication in Agronomy, with the reference “845618” comprises a selection from 50 rhizobacterial strains depending on their potential properties to prevent salt stress, testing finally the four strains that induced higher differences in wheat plants under salt stress conditions, as well as their combination.

This is a complete and interesting work, including from bacterial isolation to the test of the selected strains in salt-stressed plants, but there are some aspects of the manuscript which could be improved, including a better description of the Materials and Methods section, the study of significant differences in part of the results, and a better discussion of the results. Although I am not a native speaker and I do not consider myself an English expert, I think that there are some minor grammar mistakes which could be improved, so I suggest the revision of the language.

Consequently, I consider that the article can not be accepted in its present form. For further information of this decision, some comments are detailed in the following lines:

Abstract

In this section is not necessary to present the specific numbers obtained in the results, it would be better presenting only the provided information without the numeric data.

Introduction

Saline area is only 1.89 hectare or 1.89 million hectares.

Change “Halophytes adopt themselves” to “Halophytes adapt themselves”.

Materials and methods

“Materials and Methods” section is poorly escribed. The procedures used for each analysis should be specified, including the extraction and the quantitation methodology. In the cases of spectrophotometric analyses, the wavelengths should be specified too.

Change “2.1.Soil sampling and pysico chemical analysis” to “2.1.Soil sampling and physiochemical analysis“.

2.2: Which was the composition of “sterilized solution”? The number of isolated rhizobacteria is not a methodology, is a result, so it should not be specified here.

2.6 2.10: At which temperature and photoperiod were the plants maintained?

2.8: How was the extraction of phytohormones performed? Was any internal standard used? At which wavelengths were detected? Were them compared with commercial standards?

2.9: Why were used these primers? Are they for a specific gene detection?

2.10: At which temperature and photoperiod were the plants maintained?

2.14-2.17: Specify the extraction procedure and the wavelengths used for each analysis.

2.12: Was the water conductivity measured before the addition of leaf discs? Were any kind of controls without leaves measured?

2.15 Was any quantification performed by the use of standard curves?

Results

3.2: In the Supplementary Table 3, the siderophore production of L3 isolate is not provided.

3.3: Was a negative control without bacteria inoculation used in these experiments?

3.4: How was the source utilization pattern? Specify it in “Materials and Methods” section.

3.5: Change “The PGPR strain SR2 and SR3 produce” to “The PGPR strains SR2 and SR3 produced”. In the figure 1, include the results of the statistical analysis comparing the different bacterial strains. The data in the Figure 1 does not corresponded to the ranges described in the text. It seems that the increase of phytohormone concentration produced by SR2 and SR3 bacterial strains is only for IAA. Were any controls without bacterium evaluated?

3.6 Include the statistical analysis in figures 2 and 3 to check that the observed differences are statistically significant.

3.8 How do you explain the higher increase in electrolyte leakage in the plants inoculated with the consortium compared to control? The values were 2.29 times higher in salt-stressed plants inoculated with the consortium in comparison to non-stressed plants, whereas in non-inoculated plants, this increase was only about 1.67 times. Explain this in the discussion.

3.13 Refer table 7 in this paragraph.

Figure 4: Revise the caption, since the meaning of “Car” and “EL” is not included. Change “SODL” to “SOD”.

Discussion

This section is brief and poor. It would need a revision, including a comparison of the obtained results with previous works regarding on the deleterious effect of salt stress.

The strange phenomenon observed in electrolyte leakage (previously explained in the revision of the results) should be revised and discussed.

Author Response

ANNOTATED RESPONSE TO REVIEWER 2

No.

COMMENT

ACTION/JUSTIFICATION

Page No. /Line No.

1.      

The article entitled “Rhizobacteria isolated from saline soil induce systemic tolerance in wheat (Triticum aestivum L.) against salinity stress”, proposed for publication in Agronomy, with the reference “845618” comprises a selection from 50 rhizobacterial strains depending on their potential properties to prevent salt stress, testing finally the four strains that induced higher differences in wheat plants under salt stress conditions, as well as their combination.

This is a complete and interesting work, including from bacterial isolation to the test of the selected strains in salt-stressed plants, but there are some aspects of the manuscript which could be improved, including a better description of the Materials and Methods section, the study of significant differences in part of the results, and a better discussion of the results. Although I am not a native speaker and I do not consider myself an English expert, I think that there are some minor grammar mistakes which could be improved, so I suggest the revision of the language.

Consequently, I consider that the article can not be accepted in its present form. For further information of this decision, some comments are detailed in the following lines:

Authors are very appreciative to the worthy reviwer. The whole manuscript has been thoroughly revised and improved according to valuable sugessions.

The suggested typological, grammatical and technical correction has been incorporated in all sections of manuscript.

2.      

Abstract

In this section is not necessary to present the specific numbers obtained in the results, it would be better presenting only the provided information without the numeric data.

All numerical values of the results have been removed from the abstract as per worthy reviwers suggestion.

Page 2; line 27-32

3.      

Introduction

Saline area is only 1.89 hectare or 1.89 million hectares.

The suggested correction has been made in the Introduction section.

Page 3 ; line 45.

4.      

Introduction

Change “Halophytes adopt themselves” to “Halophytes adapt themselves”.

The suggested correction has been made in the Introduction section

Page 3; line 51

5.      

Materials and methods

Materials and Methods” section is poorly escribed. The procedures used for each analysis should be specified, including the extraction and the quantitation methodology. In the cases of spectrophotometric analyses, the wavelengths should be specified too.

The material and methods section has been thoroughly revised and details of every parameter have been added.

6.      

Materials and methods

Change “2.1.Soil sampling and pysico chemical analysis” to “2.1.Soil sampling and physiochemical analysis“.

The suggested correction has been made in section 2.1.

Page 4; line 74

7.      

Materials and methods

2.2: Which was the composition of “sterilized solution”?

The number of isolated rhizobacteria is not a methodology, is a result, so it should not be specified here.

The details of the sterilized solution have been added in Section 2.2.

Medium (LB) used for isolation of rhizobacteria has also been mentioned in Section 2.2.

 The numbers of isolated rhizobacteria have been removed from the methodology section.

Page 4; line 85

Page 4; line 82

8.      

Materials and methods

2.6 2.10: At which temperature and photoperiod were the plants maintained?

The growing conditions of the plants in the pot experiment has been added in section 2.10.

Page 5; line 110

Page 7; line 152

9.      

Materials and methods

2.8: How was the extraction of phytohormones performed? Was any internal standard used? At which wavelengths were detected? Were them compared with commercial standards?

The detailed procedure of phytohormone extraction including internal standards and wavelengths has been provided.

Yes, Commercial internal standards were used and their details have now been provided.

Page 6; line 124-134

Now its 2.6:

10.   

Materials and methods

2.9: Why were used these primers? Are they for a specific gene detection?

Universal primers have been used for amplification of the 16srRNA gene which is conserved in bacteria and widely used method for identification of bacteria.

Page 6; line 138-139

Now its 2.7:

11.   

Materials and methods

2.10: At which temperature and photoperiod were the plants maintained?

The growing conditions of the plants in the pot experiment have been added.

Page 7 line 152

Now its 2.8:

12.   

Materials and methods

2.14-2.17: Specify the extraction procedure and the wavelengths used for each analysis.

The detailed procedure of each analysis has been added to the materials and methods section.

Page 7-9; line 182-207

13.   

Materials and methods

2.12: Was the water conductivity measured before the addition of leaf discs? Were any kind of controls without leaves measured?

No, this parameter was carried out as per standard procedure by Sairam, 1994 and it doesn’t involve any measurement of electical conductivity before the addition of leaf discs.

However, a comparison with control (without any inoculation and stress) was used as a comparison.

14.   

Materials and methods

2.15 Was any quantification performed by the use of standard curves?

Quantification of protein and amino acid content was done using the standard curve of the respective standard. This information has been provided in section 2.15.

Page 7-10; line 161-225

15.   

Results

3.2: In the Supplementary Table 3, the siderophore production of L3 isolate is not provided.

The supplementary table 3 has been rectified.

16.   

Results

3.3: Was a negative control without bacteria inoculation used in these experiments?

Yes, control treatment didn’t receive any inoculation.

17.   

Results

3.4: How was the source utilization pattern? Specify it in “Materials and Methods” section.

The details of the carbon and nitrogen source (C/N) utilization pattern has been provided in the materials and methods.

Page 5; line 91-92

18.   

Results

3.5: Change “The PGPR strain SR2 and SR3 produce” to “The PGPR strains SR2 and SR3 produced”.

This suggested correction has been made 

Page 11; line 271

19.   

In the figure 1, include the results of the statistical analysis comparing the different bacterial strains. The data in the Figure 1 does not corresponded to the ranges described in the text. It seems that the increase of phytohormone concentration produced by SR2 and SR3 bacterial strains is only for IAA. Were any controls without bacterium evaluated?

We also detected the respective phytohormones production in non inoculated LB medium as control. The data  of bacterial phytohormones was normalized with control (medium without any inoculation),The values have been corrected.

Page 6: line 133-134

20.   

Results

3.6 Include the statistical analysis in figures 2 and 3 to check that the observed differences are statistically significant.

Standard deviations were calculated for all values and have been presented in the form of error bars

Page 32; line 741

Page 33; line 753

21.   

Results

3.8 How do you explain the higher increase in electrolyte leakage in the plants inoculated with the consortium compared to control? The values were 2.29 times higher in salt-stressed plants inoculated with the consortium in comparison to non-stressed plants, whereas in non-inoculated plants, this increase was only about 1.67 times. Explain this in the discussion.

The data of electrolyte leakage has been  rectified in Table 4. Moreover, details of results has been added in the results section.

Page 27; line 678

Page 16; line 408-410

22.   

Results

3.13 Refer table 7 in this paragraph.

Table # 7 has been referred to section 3.13 of results.

Page 14; line 336

23.   

Results

Figure 4: Revise the caption, since the meaning of “Car” and “EL” is not included. Change “SODL” to “SOD”.

The caption of figure 4 has been revised by including the details of  “Car” and “EL” and

changing “SODL” to “SOD”.

Page 34; line 762

Page 34; line 762-764

24.   

Discussion

This section is brief and poor. It would need a revision, including a comparison of the obtained results with previous works regarding on the deleterious effect of salt stress.

The discussion section has been rewritten and thoroughly revised as suggested by the worthy reviewer.

Page 14-17

25.   

Discussion

The strange phenomenon observed in electrolyte leakage (previously explained in the revision of the results) should be revised and discussed.

The results of electrolyte leakage have been rectified.

Page 27; line 678

Page 16; line 409-412

Page 12; line 300

Round 2

Reviewer 2 Report

In this revision, the authors have substantially improved the quality of the manuscript entitled “Rhizobacteria isolated from saline soil induce systemic tolerance in wheat (Triticum aestivum L.) against salinity stress”. However, some minor revisions are required, mainly in the new versions of the figures. Once these modifications have been implemented, the manuscript could be accepted for publication in “Agronomy”.

Graphical abstract: Change “amelioatation” to “amelioration”.

2.3 Add a brief description of these procedures.

Figure 1: In the figure caption, t-ZR has been changed to CK. This change should be added to the figure legend too.

Figure 2: Although the statistical analysis has been successfully added, some strange lines have appeared in some cases (for example, in SR3 at 2% and 6% NaCl, or SR1 8% NaCl). Please, revise it and delete these lines.

Figure 4: Due to the changes in the values of electrolyte leakage, some differences in the correlation coefficient could be derived, but the figure has not been modified. Please, revise if any change in this parameter and modify the figure if it is necessary.

Author Response

ANNOTATED RESPONSE TO REVIEWER 2 (ROUND II)

No.

COMMENT

ACTION/JUSTIFICATION

Page No. /Line No.

1.      

In this revision, the authors have substantially improved the quality of the manuscript entitled “Rhizobacteria isolated from saline soil induce systemic tolerance in wheat (Triticum aestivum L.) against salinity stress”. However, some minor revisions are required, mainly in the new versions of the figures. Once these modifications have been implemented, the manuscript could be accepted for publication in “Agronomy”.

We are thankful to the reviewer for accepting the revisions.

All suggested minor revisions are also being incorporated according to valuable suggestions.

2.      

Graphical abstract: Change “amelioatation” to “amelioration”.

Graphical abstract has been edited and word “ameliotation” has been replaced with “amelioration”

Page 2; line 18

3.      

2.3 Add a brief description of these procedures

Detailed description of these methods have been added in 2.3:

Page 5; line 96-113

4.      

Figure 1: In the figure caption, t-ZR has been changed to CK. This change should be added to the figure legend too.

t-ZR has been removed from legend as per suggestion.

Page 32; line 743

5.      

Figure 2: Although the statistical analysis has been successfully added, some strange lines have appeared in some cases (for example, in SR3 at 2% and 6% NaCl, or SR1 8% NaCl). Please, revise it and delete these lines.

This error occurred during the addition of statistical analysis, however, the discrepancy has been removed.

Page 33; line 752

6.      

Figure 4: Due to the changes in the values of electrolyte leakage, some differences in the correlation coefficient could be derived, but the figure has not been modified. Please, revise if any change in this parameter and modify the figure if it is necessary.

New co-relations were derived and new figures have been added

Page 35; line 772